# The Stress Phase Angle Measurement Using Spectral Domain Optical Coherence Tomography

**DOI:** 10.3390/s23177597

**Published:** 2023-09-01

**Authors:** Yuqian Zhao, Zhibo Zhu, Huiwen Jiang, Yao Yu, Jian Liu, Jingmin Luan, Yi Wang, Zhenhe Ma

**Affiliations:** 1School of Control Engineering, Northeastern University at Qinhuangdao, Qinhuangdao 066004, China; zhaoyanyuqian@163.com (Y.Z.); 2272364@stu.neu.edu.cn (Z.Z.); y.j.yu@outlook.com (Y.Y.); liujianneuq@gmail.com (J.L.); wangyi@neuq.edu.cn (Y.W.); 2College of Information Science and Engineering, Northeastern University, Shenyang 110819, China; 2010334@stu.neu.edu.cn; 3School of Computer and Communication Engineering, Northeastern University at Qinhuangdao, Qinhuangdao 066004, China; luanjingmin@neuq.edu.cn

**Keywords:** stress phase angle, optical coherence tomography, circumferential stress, wall shear stress

## Abstract

The stress phase angle (SPA), defined as the temporal phase angle between circumferential stress (CS) in the arterial wall and wall shear stress (WSS), is utilized to investigate the interactions between CS and WSS. SPA serves as an important parameter for the early diagnosis of cardiovascular disease. In this study, we proposed a novel method for measuring SPA using spectral domain optical coherence tomography (SD-OCT). The multi-M-mode scan strategy is adopted for interference spectrum acquisition. The phases of CS and WSS are extracted from the corresponding structural and flow velocity images of SD-OCT. The method is validated by measuring SPA in the outflow tract (OFT) of chick embryonic hearts and the common carotid artery of mice. To the best of our knowledge, this is the first time that OCT has been used for SPA measurement.

## 1. Introduction

Cardiovascular disease (CVD) continues to be a prominent global cause of mortality [1], with atherosclerosis significantly contributing to various cardiovascular conditions, including unstable angina, myocardial infarction, and stroke [2]. Hemodynamic factors play a crucial role in the development and diagnosis of atherosclerosis [3,4,5]. Gaining a comprehensive understanding of the hemodynamic forces involved can offer valuable insights for enhancing prognoses and treatments of CVDs.

Endothelial cells (ECs), which line the walls of blood vessels, are constantly exposed to two mechanical stimuli: circumferential stress (CS) caused by blood pressure and wall shear stress (WSS) resulting from blood flow [6]. ECs sense and respond to these mechanical forces through cellular processes, such as cell proliferation, apoptosis, and growth [7,8,9]. The interaction between WSS and CS plays a crucial role in arterial remodeling and atherosclerosis development. The stress phase angle (SPA), defined as the temporal phase difference between CS and WSS, provides insights into their interaction [10]. SPA reflects the degree of asynchrony between pressure and flow waveforms, capturing the combined effects of mechanical forces on the vessel wall and hemodynamic factors on endothelial cells. Recent research has revealed that SPA not only influences vascular function but also holds promise as an early diagnostic criterion for CVDs, owing to its role in regulating the release of vasoactive molecules such as nitric oxide (NO), endothelin-1 (ET-1), and prostacyclin (PGI2) [11,12,13,14]. Studies have observed a correlation between blood vessel wall elasticity and SPA [15], with plaques predominantly found in regions exhibiting large negative SPA values [16]. Accurate measurement of SPA is essential to investigate the role of SPA in CVDs.

Several technologies have been developed to measure the CS of the vessel wall and the shear stress of blood flow. Wise et al. measured velocity maps in the ventricles and ventricular walls to show blood flow and wall motion patterns in rats with an MRI technique [16]. However, MRI has limitations in terms of long imaging time and the inability to provide real-time measurements. Moreover, to explore the role of the stress phase in blood vessels, computer simulations have also been applied to study the relationship between SPA and other hemodynamic factors in blood vessels [6,17]. Ultrasound technology enables the measurement of vessel wall motion and blood flow. Luo et al. employed the speckle tracking method to simultaneously measure arterial wall axial motion and blood flow velocity, but this technique requires a high frame rate [18]. Xu and Niu et al. exploited ultrasonic biomechanics (UBM) based on ultrasonic B-mode contrast images to calculate the SPA [19,20]. The UBM method was validated through experiments using silicon tubes and PVA-c phantoms. However, the relatively low spatial resolution of the ultrasound can impact the accuracy of the measurement. Therefore, a technique capable of accurately assessing vessel wall motion and blood flow patterns would be beneficial for measuring SPA.

Optical coherence tomography (OCT), known for its high spatial and temporal resolution, has been widely employed in hemodynamics research. OCT enables the acquisition of structural and flow velocity images of the sample with a single acquisition. In 1997, Chen et al. proposed an optical Doppler tomography system to measure flow velocity distribution [21]. Using phase-resolved Doppler OCT, Ma et al. measured absolute blood flow velocity by utilizing vascular structural images obtained through OCT [22]. Additionally, 4D structural and Doppler OCT imaging was employed to measure WSS in the quail embryonic heart in vivo [23]. In arteries, the pulsatile blood flow leads to changes in CS, causing corresponding deformation of the blood vessel wall. Ma et al. [24] estimated vessel wall strain using OCT structural images. Li et al. measured the movement velocity on the myocardial wall to extract the radial strain and strain rate of the blood vessel [25]. However, to date, OCT has not been utilized for measuring SPA.

In this study, SPA was measured using a special domain OCT (SD-OCT) system. The CS was extracted from the structural images, while the WSS was calculated from the Doppler flow images. By applying the proposed method, the SPA of the chicken embryonic heart outflow tract (OFT) and mouse carotid artery were measured in vivo. The results highlight the potential of OCT as a valuable imaging technique for SPA assessment.

## 2. Methods

### 2.1. SD-OCT System

Figure 1 illustrates the schematic of the OCT system adopted in this study, which is similar to the previously described [26]. This OCT system uses a broadband infrared superluminescent diode (SLD) with a 1310 nm central wavelength and 52 nm bandwidth, providing ~14 μm axial resolution in air. The emitted light passes through an optical circulator and couples into a fiber-based Michelson interferometer (50:50). In the sample arm, a two-dimensional (2D) scan was realized by an X-Y galvanometer scanner. The probing beam is focused onto the sample by an objective lens (f = 50 mm), offering a lateral resolution of ~16 μm. The optical interference signal between the light backscattering from the sample and reference arms was routed into a homemade spectrometer, which adopts an infrared achromatic doublet collimating lens (f = 50 mm), an 1145 line/mm transmission grating, a Fourier lens (f = 100 mm), and a 1024-element infrared InGaAs line-scan camera with a 92 kHz maximum scan rate. The spectral resolution is ~0.128 nm, producing an imaging depth of ~2.95 mm in air. Polarization controllers in the system were used to maximize the interference between the sample and reference arm. In addition, a 633 nm visible light from a laser diode was coupled into the interferometer to facilitate the positioning of the probe beam on the sample.

### 2.2. Structure and Doppler Flow Imaging with SD-OCT

SD-OCT technology is based on spectral interference; that is, the interference spectrum is recorded by a spectrometer [23]:(1)$$I\left(k\right)=2S\left(k\right)E_{R}\int_{-\infty }^{\infty }a\left(z\right)\cos\left(2\mathrm{knz}\right)\mathrm{dz}$$
where k is the wave number, n is the refractive index of the tissue, z is the sample depth, S(k) is the spectral density function of the light source, E_R_ is the reference light amplitude, and a(z) is the amplitude of backscattering light from different depth of the sample. Depth-resolved information of the sample can be extracted by fast Fourier transformation (FFT) of the interference spectrum (I(k)). The FFT result is a complex sequence that contains amplitude and phase information. The amplitude is used to construct a structural image of the sample, while the phase information can be utilized to calculate Doppler flow velocity.

For OCT imaging, a B-scan image is composed of multi-A-lines which are acquired sequentially. During adjacent two A-lines, moving particles in the sample can introduce a phase difference [23]:(2)$$\Delta \varphi \left(z\right)=\frac{\;4\pi n\eta V\left(z\right)\cos\theta }{\lambda }$$
where η represents the time interval between two adjacent A-scans. V(z) is the flow velocity distribution; λ is the central wavelength of the light source; n is the refractive index of the tissue; θ is the angle between the velocity vector and the probe light, i.e., the Doppler angle. From Equation (2), the absolute flow velocity V(z) is derived:(3)$$V\left(z\right)=\frac{\theta \Delta \varphi \left(z\right)}{4\pi n\eta \cos\theta }$$

Figure 2 illustrates typical OCT images of a polyethylene capillary phantom. The capillary, with an inner diameter of 0.5 mm and an outer diameter of 0.8 mm, was filled using a microsyringe pump (Genie Touch, Kent Scientific, CT, USA). The capillary was immersed in a 5% agar solution to reduce the structural deformation caused by the optical path length difference. Figure 2a displays the structural image, and Figure 2b shows the flow velocity image. Both structural and flow velocity images were obtained from the same raw data using SD-OCT.

### 2.3. SPA Extraction

The SPA is a parameter primarily measured in arteries, which consist of a three-layered membrane structure: tunica intima, tunica media, and tunica adventitia. The intima is predominantly composed of a single layer of endothelial cells that directly interface with the blood, playing a role in regulating the inner diameter of blood vessels by sensing blood flow [27]. The flow of blood in the large arteries is typically pulsatile and unidirectional. Figure 3 provides a schematic representation of the SPA. The blood flow exerts positive pressure (i.e., blood pressure (P)) perpendicular to the vessel wall, causing the deformation of the vessel (i.e., circumferential strain). Simultaneously, due to blood viscosity, the WSS is generated at the point of contact between the vessel wall and the blood flow (V). These two forces act concurrently and directly influence the morphology, structure, and function of the endothelium, thereby contributing to the formation and development of vascular disease [28,29]. The SPA, defined as the phase angle between the CS (σ(t)) and the WSS (τ(t)), reflects the asynchronous nature of the pressure and fluid waveforms:(4)$$\mathrm{SPA}=\varphi \left[\sigma \left(t\right)\right]-\varphi \left[\tau \left(t\right)\right]$$

#### 2.3.1. Phase of CS Extraction Based on Blood Vessel Deformation

The extraction of the phase angle for the circumferential strain is based on blood vessel deformation. According to the theory of elastic mechanics, the CS of a thin-walled elastic tube can be expressed as follows [30]:(5)$$\sigma =E\frac{\Delta C}{C_{0}}$$
where C_0_ is the minimum circumference of the blood vessel; ∆C is the variation of vessel circumference introduced by pulsatile blood flow; E is Young’s modulus. For a fixed position of the blood vessels, both C_0_ and E are constants. Assuming the cross-section of the blood vessel is circular, the CS is proportional to the variation of the vessel diameter (D(t)), implying that vessel diameter variation is in phase with wall pressure [31]. Thus, Equation (4) can be rewritten as:(6)$$\mathrm{SPA}=\varphi \left[D\left(t\right)\right]-\varphi \left[\tau \left(t\right)\right]$$

#### 2.3.2. WSS Calculation in Blood Vessel

WSS is a frictional force parallel to the inner wall of the vessel, opposite to the direction of blood flow. The blood is modeled as an inelastic straight and laminar Newtonian fluid. The WSS (τ) can be expressed as the product of the viscosity (μ = constant) and velocity gradient [32]:(7)$$\tau =\mu {\left.\frac{\partial V\left(r\right)}{\partial r}\right|}_{r=R}$$
where V is the absolute flow velocity, r is the radius of a certain point in the blood vessel, and $\frac{\partial V\left(r\right)}{\partial r}=\dot{\gamma }$ represents the wall shear rate (WSR). According to Equation (7), absolute flow velocity calculates the WSS. On the other hand, OCT can only provide a velocity component parallel to the probing beam. A Doppler angle is required to achieve absolute flow velocity (Equation (3)). Doppler angle obtaining increases the difficulty of the SPA measurement. For a certain SPA measurement on the blood vessel, the Doppler angle remains constant during the OCT imaging. Thus, cosθ is a constant (Equation (3)) and affects the amplitude calculation of the WSS but does not affect the phase. Fortunately, the SPA is a parameter concerning the CS and WSS’s phase relationship. We can achieve the phase of the WSS with the OCT-provided flow velocity component information and, needless to consider the Doppler angle. Similarly, μ is a constant, and the WSS is proportional (in phase) to the WSR. Thus, the phase of the WSR can be utilized for the SPA calculation:(8)$$\mathrm{SPA}=\varphi \left[D\left(t\right)\right]-\varphi \left[\dot{\gamma }\left(t\right)\right]$$

Eventually, the SPA can be achieved by extracting the phase of vessel diameter variation and the WSR of blood flow.

#### 2.3.3. OCT Multi-M-Mode Scan for SPA Extraction

To measure the SPA, the OCT requires providing both structural (for CS) and flow (for WSS) information. In our study, we designed a multi-M-mode scan in the OCT acquisition process. During the acquisition, the probing beam was focused on the target blood vessel and kept stationary (see Figure 4a). The line scan camera of the OCT operated at 65,000 Hz in a free-run mode. Each M-mode scan consisted of 50 A-lines. Figure 4b shows a structural image of a single M-mode scan using a chick embryo heart OFT as the sample, and Figure 4d shows the corresponding flow image. In total, the OCT acquired 1000 M-mode scans (f(i), i = 1, 2, … N) with a 400 Hz frame rate. The pure acquisition time of each M-mode scan was ~0.8 ms, significantly smaller than the heartbeat period (generally more than 100 ms). Thus, the vascular deformation during each M-mode acquisition was negligible. We averaged A-lines within a single M-mode scan and obtained one A-line to build the final M-mode image (Figure 4c structure and Figure 4e flow). The final M-mode image consisted of 1000 averaged A-lines (corresponding to 1000 M-mode scans).

#### 2.3.4. SPA Extraction from M-Mode Images

According to the previous analysis, blood vessel diameter variation can be used for the phase of the CS extraction. For the achieved structure M-mode image, the variation of blood vessel diameter can be assumed to be in phase with the fluctuation of blood vessel boundary since the M-mode image was acquired at the middle of the blood vessel (Figure 4a). Boundary segmentation was performed on the M-mode structural image. Firstly, grayscale transformation was applied for image enhancement, and a Gaussian high-pass filter was used to highlight the image edge. Secondly, the boundaries were enhanced by the 4 × 1 descending gradient template ([1, 1, −1, −1]). Lastly, morphological operations (open and closed operations) were employed to extract boundaries. Figure 5a illustrates the result of boundary segmentation in the structure M-mode image (solid yellow line) of chicken embryo OFT. The fluctuation of the blood vessel boundary corresponds to the variation of the CS. Based on the segmented boundary, the WSR of blood can be calculated in a Doppler flow M-mode image (Figure 5b), and the result is shown in Figure 5c. Thus, the SPA of blood vessels can be achieved according to Equation (8).

## 3. Experiment and Result

### 3.1. SPA Measurement of Chick Embryonic Heart OFT

In this study, the SPA of chick embryonic heart OFT was measured in vivo. Fertilized White Leghorn chicken eggs were incubated with the blunt end facing up at a temperature of 38 °C and 80% humidity until reaching stage HH18 (~3 days; Hamburger and Hamilton, 1992). According to the regulations of the Animal Ethics and Administrative Council of Northeastern University, chicken embryos are not considered vertebrates. Nevertheless, we made every effort to minimize the number of embryos needed. A portion of the eggshell and chorionic membrane was carefully removed, and the eggs were placed in the sample arm of the OCT system to acquire data. To maintain a constant temperature of 37.5 °C, a heating blanket was employed.

A multi-M-mode scan was conducted on the OFT of a chick at five different positions (indicated by dashed lines in Figure 6a). In Figure 6b–f, the red lines represent the fluctuation of the vessel boundary, which corresponds to the CS, while the blue lines depict the WSR profiles, which are proportional to the WSS. Thus, the SPA can be measured by comparing the phase difference between the two curves. The SPA values at the five different positions are listed in Table 1. At each position, the SPAs were measured over three consecutive cardiac cycles, and the values of the SPA remained stable, indicating that the proposed method was effective. However, it is important to note that there are certain variations observed when comparing the SPAs measured at different positions. These variations can be attributed to the irregular shape and varying geometries of the vessels in the OFT, which is the distal region of the embryonic heart connecting the ventricle with the arterial system [33]. Additionally, the motion of the OFT is complex, involving not only dilation and contraction but also bending, torsion, and irregular vessel movements, all of which can influence the SPA values.

### 3.2. In Vivo Mouse Common Carotid Artery

Experiments were conducted on adult C57BL/6 mice weighing between 20 and 30 g. All procedures were performed in accordance with the regulations of the Animal Ethics and Administrative Council of Northeastern University. All possible efforts were made to minimize animal suffering and reduce the number of animals used. Surgical anesthesia was induced with sodium pentobarbital (3%, 5 mg/100 g, IP). The anesthetized mice were securely positioned on a stereotaxic device (ST-5ND-C) with ear bars and a clamping device. The fur on the neck was carefully shaved, and the skin was cleaned with a 0.9% sodium chloride physiological solution. A longitudinal incision was made in the neck skin, allowing for the exposure of the underlying tissues and subsequent visualization of the carotid artery.

The SPA of the mouse common carotid artery was measured by SD-OCT using the proposed method. Figure 7a shows the M-mode structural image of the mouse carotid artery, with the red line representing the extracted vessel boundary. Figure 7b displays the corresponding M-mode Doppler flow image, where the blue areas indicate phase wrapping caused by flow velocities exceeding the measurable range of the SD-OCT system. To address this, the phase-unwrapped image is presented in Figure 7c [34]. Thus, the SPA can be achieved by comparing the phase difference between the CS and WSR (Figure 7d). The SPA values obtained from three cardiac cycles were achieved (−82.11 ± 1.61°).

## 4. Discussion

In this study, we proposed a method based on the SD-OCT to measure the SPA of blood vessels in animal models. Because of the influence on ECs gene expression, protein, and metabolite secretion, the CS and WSS are well known for their role in the localization of atherosclerosis [35]. The SPA represents the correlation between these two mechanical forces [10] and has recently been confirmed to be used as an early diagnostic parameter of coronary atherosclerosis [36]. However, CS and WSS measurements require different techniques generally, making SPA measurement challenging. OCT is capable of providing structural and flow velocity images by detecting the interference spectrum between the reference arm and sample arm, offering the potential for SPA measurement. While the phase of the WSS can be directly extracted from the flow velocity image, the CS cannot be measured directly by OCT. Fortunately, assuming that the blood vessel behaves as a thin-walled elastic tube, the vessel deformation is in phase with the CS, and this deformation can be extracted from the structural OCT image. Thus, the SPA measurement is realized using a purely OCT-based technique. To the best of our knowledge, this is the first time that OCT has been used to measure SPA.

For the proposed method, the flow velocity image is used for the WSS extraction. In the SD-OCT, the phase difference between adjacent two A-lines is calculated to obtain flow velocity (Δφ in Equation (3)). However, the calculated phase is limited to a range of −π to +π, and phase wrapping occurs when the flow velocity exceeds a certain threshold value. According to Equation (3), the threshold value can be increased by decreasing the time interval (τ) between A-lines. Therefore, a high A-line acquisition speed is required to avoid phase wrapping. On the other hand, the cardiac rate of the animal model is typically a few Hertz. High line scan speeds (tens of thousands of Hertz) may cause data redundancy. To address this dilemma, we designed a multi-M-mode scan approach. During each M-mode scan, the line scan speed is set at 65,000 lines/s, effectively reducing the occurrence of phase wrapping effectively. To prevent data over-acquisition, each M-mode scan only contains 50 A-lines, with a certain time interval set between M-mode scans. As a result, the imaging speed of the M-mode scan is 400 frames/s, which is still sufficient for cardiac phase extraction. For each M-mode scan, the assumption is made that the flow velocity of blood remains constant since the acquisition time is short (~8 ms). Thus, we can average these A-lines to increase the SNR of the signal.

Although employing a high A-line acquisition speed can help reduce phase wrapping, it may not completely eliminate it, particularly when the blood flow velocity is very high (Figure 7b). Increasing the Doppler angle can alleviate the issue of phase wrapping to some extent (Equation (3)). However, it should be noted that the accuracy of the velocity measurement deteriorates when the Doppler angle approaches π/2 [37]. Generally, blood flow in vessels exhibits quasi-laminar flow, i.e., where the flow velocity gradually increases from the border to the center. Thus, phase wrapping tends to be more pronounced in the central area of the blood vessel. Fortunately, the WSS is calculated at the inner border of the blood vessel wall. Avoiding phase wrapping at the border is comparatively easier to achieve, which addresses this specific requirement.

Though previous studies reveal the potential of the SPA for early diagnosis of CVDs, there are no effective methods for SPA measurement. Currently, ultrasound imaging [20] and computational fluid dynamics (CFD) [36] have been used in the measurement of SPA. Ultrasound imaging measures the arterial strain based on ultrasonic B-mode scanning and simultaneously obtains flow velocity using echo particle image velocimetry (EPIV). The comparatively low resolution of ultrasound imaging affects the accuracy of the SPA measurement. CFD utilized various imaging techniques (such as MRI) and simulation to obtain SPA. The reliability of CFD requires further validation. Thus, there are few clinical applications of SPA so far. In contrast, OCT offers several advantages, including fast acquisition, non-contact measurement, and high resolution. OCT can record structural deformations and flow velocity changes in the vessel simultaneously, through which the corresponding CS and WSS can be obtained. Therefore, the OCT gives the potential for accurate SPA measurement. However, the primary limitation of OCT measurements is its limited penetration depth (around 1–2 mm in turbid media), restricting the application of the SPA measurement to animal models. Although swept-source OCT exhibits better penetration performance than the SD-OCT system employed in this study, it is still insufficient for clinical SPA measurement. The clinical implementation of OCT-based SPA measurement relies not only on advancements in OCT technology but also on the development of other related technologies, such as optical clearing, to extend the penetration depth of OCT imaging.

## 5. Conclusions

In this study, we have proposed a method to measure the SPA using SD-OCT. A multi-M-mode scan approach with SD-OCT was performed to address the challenges of phase wrapping and data redundancy. The phase of the WSS was extracted from the flow velocity image obtained through OCT. The phase of the CS was determined by calculating blood vessel deformation in the structural image of the OCT. Using the proposed method; we successfully measured SPAs of chick embryonic OFT and common carotid artery of mice. The experimental results demonstrate the potential of OCT as a valuable tool for SPA measurement in animal models. This opens up opportunities for further research in the field of cardiovascular biomechanics and the early detection of cardiovascular diseases.

## Figures and Tables

**Figure 1 sensors-23-07597-f001:**
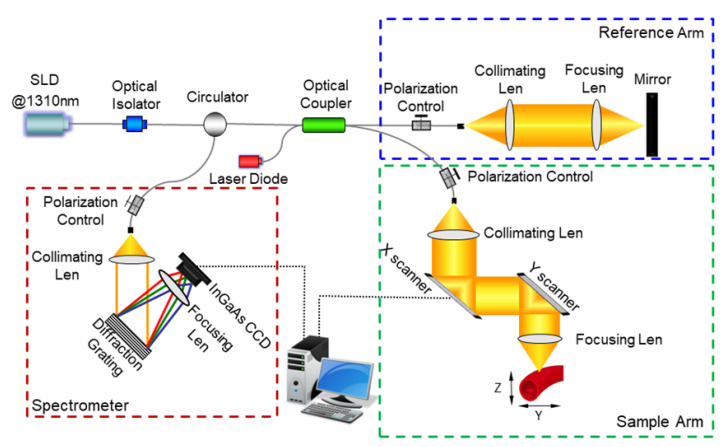
Schematic of a spectral-domain OCT system. Red box: spectrometer; blue box: reference arm; green box: sample arm.

**Figure 2 sensors-23-07597-f002:**
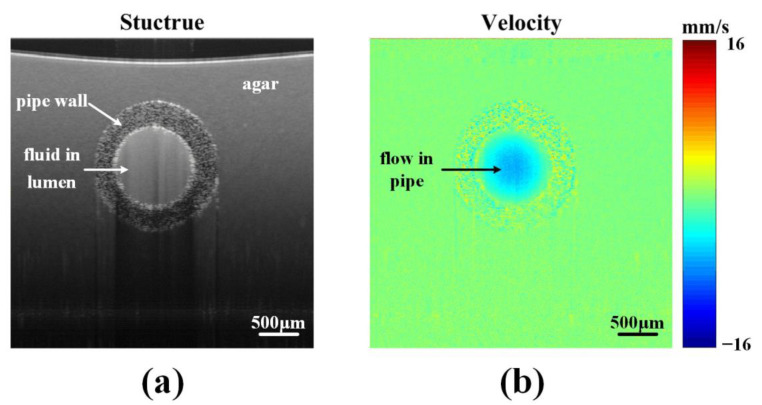
OCT images of a polyethylene pipe phantom. (**a**) The cross-sectional structural image. (**b**) The flow velocity image.

**Figure 3 sensors-23-07597-f003:**
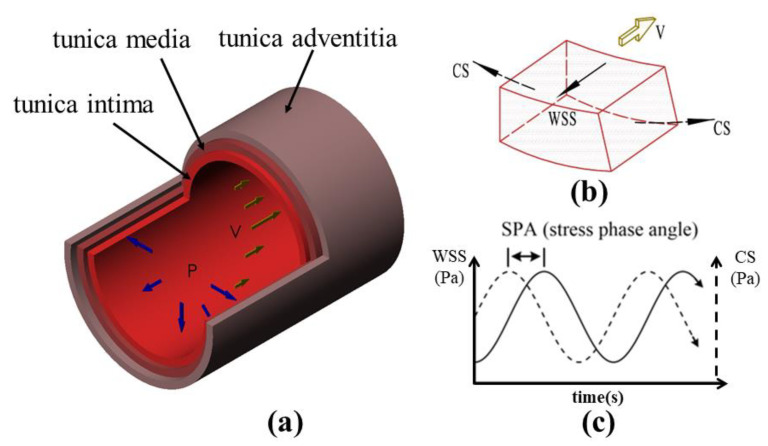
Schematic of the stress phase angle (SPA). (**a**) Three-layer structure of blood vessel wall. (**b**) Stress states of the tunica intima element. (**c**) CS and WSS change over time and SPA calculation. Blood pressure (P) induces CS, and blood flow (V) causes WSS.

**Figure 4 sensors-23-07597-f004:**
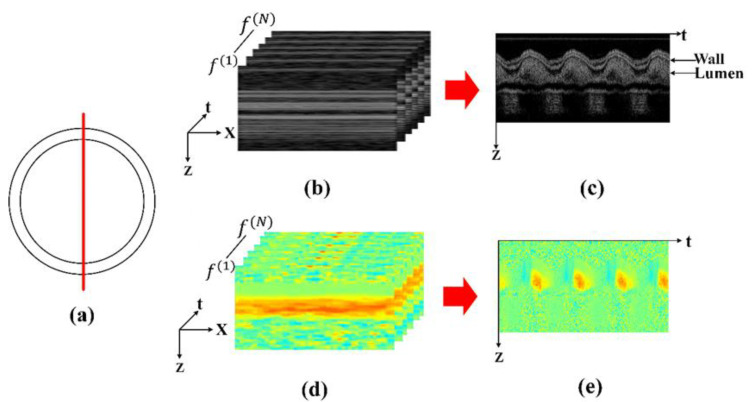
M-mode extraction process. (**a**) Position of the M-mode scan (red line). (**b**) Structure images of the multi-M-mode scan. (**c**) Final structure M-mode image extracted from the single M-mode scan A-lines average. (**d**) The corresponding Doppler flow image of (**b**). (**e**) Final Doppler flow M-mode image corresponding to (**c**).

**Figure 5 sensors-23-07597-f005:**
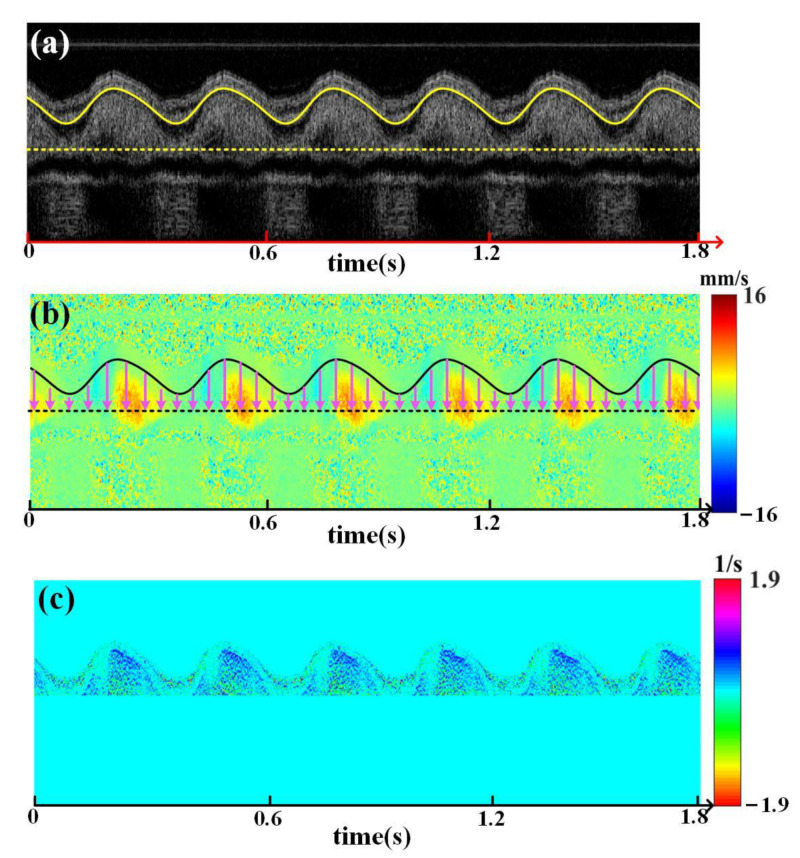
(**a**) Segmented boundary of chick embryo OFT in the M-mode image (solid yellow line); (**b**) M-mode image of blood flow corresponding to (**a**); (**c**) calculated wall shear rate (WSR) based on (**a**,**b**).

**Figure 6 sensors-23-07597-f006:**
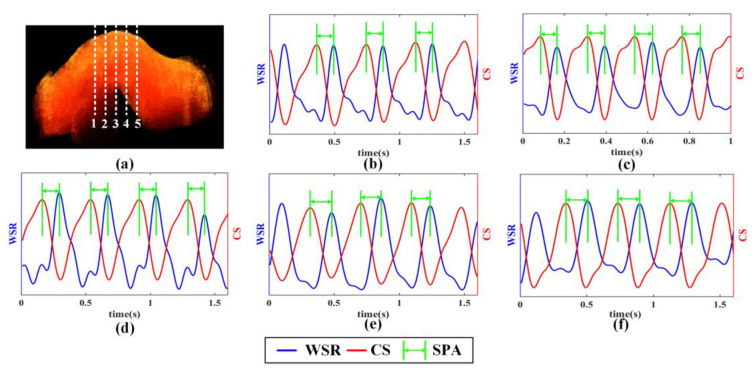
SPA measurement at different positions of chick embryo OFT. (**a**) Positions selection on the OFT to measure the SPA (white dashed line 1–5: five positions for SPA measurement); (**b**–**f**) Curves of the WSR and CS for the SPA extraction.

**Figure 7 sensors-23-07597-f007:**
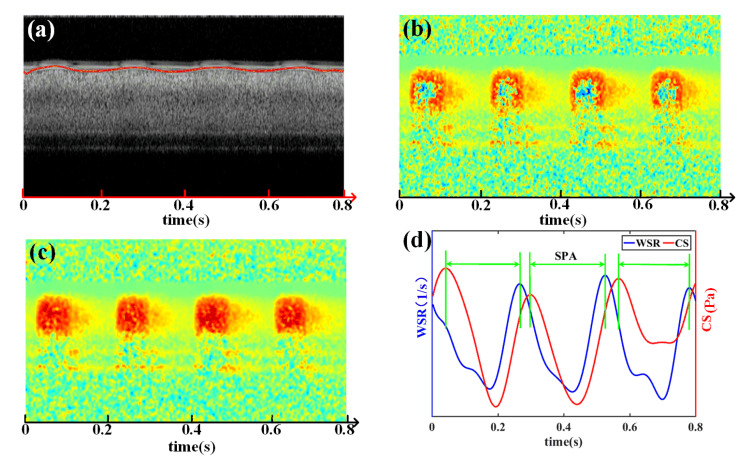
(**a**) M-mode structural image of mouse common carotid artery. Red line: Boundary of the vessel wall. (**b**) M-mode flow velocity image corresponding to (**a**). (**c**) Phase unwrapping result of (**b**); (**d**) SPA measurement of three cardiac cycles.

**Table 1 sensors-23-07597-t001:** The value of the SPA in chicken embryonic OFT.

Position	1	2	3	4	5
SPA	−46.58	−51.84	−47.74	−57.20	−59.11
	−45.97	−49.54	−47.16	−56.12	−59.00
	−45.72	−50.69	−47.23	−56.16	−59.04
	−46.09 ± 0.13	−50.69 ± 0.88	−47.38 ± 0.07	−56.50 ± 0.25	−59.05 ± 0.002

## Data Availability

The data that support the findings of this study are available from the corresponding author upon reasonable request.

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
