# Peer review of "The Stress Phase Angle Measurement Using Spectral Domain Optical Coherence Tomography"

_sensors, 2023, doi:10.3390/s23177597_

Round 1

Reviewer 1 Report

The authors propose a method to measure SPA using spectral 13 domain optical coherence tomography (SD-OCT). Multi-M-mode scan strategy is adopted for inter-14 ference spectrum acquisition. The phases of CS and WSS are extracted from corresponding struc-15 tural and flow velocity images of SD-OCT. The results highlight the potential 75 of OCT as a valuable imaging technique for SPA assessment. The method sounds a little interesting. However, the paper needs a minor revision before publication.

1 Line 63 “Using phase- resolved Doppler OCT……” is Using phase-resolved Doppler OCT?

2 Line 144 and 157: Why two phrases become the independent paragraphs? I don’t think it is reasonable in this form.

Author Response

We would like to express our great gratitude to the reviewers for their constructive comments and suggestions. Each reviewer's concern has been carefully addressed and responded to point-by-point in the following responses and revisions. We have made our best efforts to revise the manuscript, and we hope these changes can lead to the acceptance of the manuscript.

NOTE: The black and italic fonts are the reviewer’s Comments, the blue fonts are responses to the comments, the red fonts are the modifications made in the article, the black fonts are the unchanged text in the article, and the green are the references.

Reviewer 2 Report

The paper applies OCT to determine the SPA using temporal algorithms. The idea is interesting; however, there are some comments about the results presentation and the experiment.

1. The authors mention that the optical setup is similar to the one in ref 26. However, more detailed information about this setup is needed. There is no mention of the polarization control (purpose), lens (specs), grating (specs), camera, and so on. This information is helpful for readers initiating this topic.

2. Equations 1 and 2 require a reference unless they are introduced by the authors, where they need to be described in detail.

3. Figure 2 contrast is poor, making it hard to observe the sample. Besides, axes with units are mandatory to understand the object's scale. Besides, the flow direction is perpendicular to the observation direction? How do the authors retrieve Doppler signals with such an angle?

4. Figure 3 needs to be identified as (a), (b), and (c) parts. The plot needs axes units; the transparent box indicating WSS and CS is not easy to see; please improve it. The artery schematic shows a variable V which is not described in the text.

5. Equation 7 is coming from…? Please include the reference.

6. The B scan is centered in the artery (figure 4a); how critical is this position if the involuntary movement of the artery is present?

7. Not sure if the OFT term is defined before it is mentioned.

8. Figure 5 also needs axes units; Figure 5c is not in the figure caption. The blue-to-red color bars are confusing; are they the same colors but with different units? (5b and 5c). What do the units s-1 imply?

9. Figure 7 also requires axes units.

10. Results of SPA are compared with a known sample or database? How accurate is the method? More information about this point is needed as it is the main objective of the work.

Author Response

(The authors gave the same response as above.)

Reviewer 3 Report

The article is well-written and discusses a novel method to measure SPA using SDOCT. It has a detailed report of the technical aspects of the study.

Here are a few minor changes suggested.

Specific comments:

Line 37: effects of wall mechanical – can this be rephrased as “effects of mechanical forces on the vessel wall” or something like that?

Line 37: should it be ‘hydro’dynamic factors or ‘hemo’dynamic factors?

Line 58: “accurately” word is missing. There are methods to measure the wall motion and blood flow, but due to limitations of the measuring technology, these measurements are not accurate, so there is a need for a technique assessing these parameters accurately. 

The quality of the language used in the article is good. 

Author Response

We would like to express our great gratitude to the reviewers for their constructive comments and suggestions. Reviewer's concern has been carefully addressed and responded to point-by-point in the following responses and revisions. We have made our best efforts to revise the manuscript, and we hope these changes can lead to the acceptance of the manuscript.

NOTE: The black and italic fonts are the reviewer’s Comments, the blue fonts are responses to the comments, the red fonts are the modifications made in the article, the black fonts are the unchanged text in the article, and the green are the references.
